# Tryptophan Intake and Metabolism in Older Adults with Mood Disorders

**DOI:** 10.3390/nu12103183

**Published:** 2020-10-18

**Authors:** Cezary Chojnacki, Tomasz Popławski, Jan Chojnacki, Michał Fila, Paulina Konrad, Janusz Blasiak

**Affiliations:** 1Department of Clinical Nutrition and Gastroenterological Diagnostics, Medical University of Lodz, 90-647 Lodz, Poland; jan.chojnacki@umed.lodz.pl (J.C.); paulina.konrad@umed.lodz.pl (P.K.); 2Department of Molecular Genetics, Faculty of Biology and Environmental Protection, University of Lodz, 90-236 Lodz, Poland; tomasz.poplawski@biol.uni.lodz.pl (T.P.); janusz.blasiak@biol.uni.lodz.pl (J.B.); 3Department of Neurology, Polish Mother Memorial Hospital Research Institute, 93-338 Lodz, Poland; michalfila@poczta.onet.pl

**Keywords:** tryptophan, diet in the elderly, depression, mood disorders, serotonin and kynurenine pathways of tryptophan metabolism

## Abstract

The role of serotonin in the pathogenesis of depression is well-documented, while the involvement of other tryptophan (TRP) metabolites generated in the kynurenine pathway is less known. The aim of this study was to assess the intake and metabolism of TRP in elderly patients with mood disorders. Ninety subjects in three groups, 30 subjects each, were enrolled in this study: controls (healthy young adults, group I) and elderly individuals without (group II) or with (group III) symptoms of mild and moderate depression, as assessed by the Hamilton Depression Rating Scale (HAM-D) and further referred to as mood disorders. The average TRP intake was evaluated with the nutrition calculator. Urinary levels of TRP, 5-hydroxyindoleacetic acid (5-HIAA), L-kynurenine (KYN), kynurenic acid (KynA), xanthurenic acid (XA), and quinolinic acid (QA) were determined by liquid chromatography with tandem mass spectrometry and related to creatinine level. The average daily intake of TRP was significantly lower in group III than the remaining two groups, but group III was also characterized by higher urinary levels of KYN, KynA, XA, and QA as compared with younger adult individuals and elderly patients without mood disorders. Therefore, mild and moderate depression in the elderly may be associated with a lower intake of TRP and changes in its kynurenine metabolic pathway, which suggests a potential dietary TRP-based intervention in this group of patients.

## 1. Introduction

The process of aging influences structural, functional, biochemical, molecular, and genetic characteristics in many cells and tissues, affecting several individual somatic and behavioral features [1]. The elderly often have sleep disorders and the pathogenesis of these syndromes can be related to various factors, including the use of caffeine, tobacco, and alcohol, sleep habits, and comorbid diseases [2]. Older adults also often suffer from mood disorders and possible causes include sociopsychological factors, such as weakening of family and social ties and feeling of loneliness [3]. However, aging may affect all internal organs, including the brain. Multi-causal, accumulating organic damage could also contribute to behavioral and mood disorders and a range of affective problems in elderly patients. Furthermore, the activity of digestive and endocrine glands decreases with aging [4].

It is commonly agreed that a decreased serotonergic function is involved in the onset and progression of depression [5]. However, serotonin is involved in many physiological functions and behavioral processes, including mood, appetite, sleep, activity, suicide, sexual behavior, and cognition; and all of them can be affected in depression [6]. The great majority of serotonin (up to 90%) is produced and metabolized in the digestive tract and the rest in the nervous system [7]. It is synthesized in the gastrointestinal tract by enterochromaffin cells, mast cells, lymphocytes, and intestinal bacteria with the involvement of tryptophan (TRP) hydroxylase 1 (TPH-1) and in the central nervous system, with the involvement of TPH-2. Approximately 5% of TRP is metabolized in the serotonin pathway and the rest in the kynurenine pathway with the involvement of indole 2,3-dioxygenases (IDO-1 and IDO-2) [8]. Although TPH and IDO enzymes compete for access to tryptophan, healthy subjects with a balanced diet have the amount of TRP high enough for both enzymes [9]. Whereas usual TRP intake is approximately 900–1000 mg daily, the recommended daily allowance for adults is projected between 250 mg/day and 425 mg/day, which corresponds to a dietary intake of 3.5–6.0 mg/kg of body weight per day [10]. Another enzyme, tryptophan 2,3-dioxygenase (TDO), is found mainly in the liver, but also in other organs and metabolizes TRP to carbon dioxide, water, and ATP, limiting its level in the blood.

Many factors affect TRP metabolism. The expression of TPH-1 in the digestive tract is potentiated by some nutrients, bacteria, and pro-inflammatory cytokines and reduced by stress hormones [11]. Cytokines, such as interferon-α (IFN-α), IFN-β, tumor necrosis factor-α (TNF-α), and IFN-γ may upregulate IDO expression [12]. Kynurenine metabolites, including 3-hydroxy-kynurenine (3-OH-KYN) and quinolinic acid (QA), may affect brain function [13]. 3-OH-KYN may induce oxidative stress by increased production of reactive oxygen species (ROS), and QA may overstimulate hippocampal *N*-methyl-D-aspartate (NMDA) receptors, leading to apoptosis and hippocampal atrophy and both these effects have been associated with depression [12].

The intake of different amounts of TRP may differentially affect its metabolism and older adults may be characterized by a diverse consumption of this amino acid than average. Moreover, in the elderly, the activity of the enzymes metabolizing TRP may change in comparison to younger individuals. These features may lead to serious consequences resulting from different TRP metabolic pathways in older adults than others. In the present work, we investigated the association of the occurrence of mood disorders with tryptophan intake and metabolism in the elderly in comparison with younger and older adults without such disorders.

## 2. Materials and Methods

### 2.1. Patients

Ninety subjects, 60 women and 30 men, aged 36–85 years, were enrolled in this study. The study was performed in 2016–2020 in the Department of Clinical Nutrition and Gastroenterological Diagnostics and in the Department of Gastroenterology, Medical University of Lodz, Lodz, Poland. Initially, each subject was assessed for mental condition using the Hamilton Depression Rating Scale (HAM-D). The following score criteria were adopted: 0–7, no mental disorder; 8–12, mild depression; 13–18, moderate depression; 19–29, severe depression; over 30, very severe depression. Then, three groups were selected: group I, subjects without mood disorders and other ailments, aged 36–52 years; group II, subjects without mood disorders, aged 65–82 years; group III, patients with symptoms of mild and moderate depression (depressive mood disorders), aged 69–85 years.

In order to determine the occurrence of somatic diseases, clinical tests were carried out in all subjects to assess the condition of the circulatory, digestive, and nervous systems. Exclusion criteria were: circulatory or respiratory failure, advanced diabetes, liver diseases, renal failure, inflammatory bowel diseases, cancer, and use of psychotropic and sleeping pills.

### 2.2. Nutrition Procedures

All individuals were recommended to record the type and quantity of products consumed per day for 21 days prior to investigations in the nutrition diary. The average TRP intake was then calculated using the nutrition calculator with the application Kcalmar.pro—Premium (Hermex, Lublin, Poland). After 21 days, biochemical testing of blood and urine was performed.

### 2.3. Laboratory Tests

The following fasting blood tests were performed: blood cell count, C-reactive protein, glucose, bilirubin, urea, creatinine, profile of lipids, thyroid-stimulating hormone, free thyroxine, free triiodothyronine, vitamins D3 and B12, alanine and asparagine aminotransferases, gamma-glutamyl transpeptidase, alkaline phosphatase, amylase, and lipase. Urine samples for the analysis of TRP metabolites were collected in the morning on an empty stomach into a special container with a solution of 0.1% hydrochloric acid as a stabilizer. Using liquid chromatography with tandem mass spectrometry (LC-MS/MS, Ganzimmun Diagnostics AG, Mainz, Germany), we determined the concentration of L-tryptophan and its following metabolites: 5-hydroxyindoleacetic acid (5-HIAA), L-kynurenine (KYN), kynurenic acid (KynA), xanthurenic acid (XA), and quinolinic acid (QA). The levels of these metabolites were expressed in mg per gram of creatinine (mg/gCr). The ratios of the levels of 5-hydroxyindoleacetic acid and tryptophan as well as kynurenine and tryptophan were also calculated. The 5-HIAA/TRP ratio was considered as an indicator of TPH-1 activity and the KYN/TRP ratio reflected the activity of DOI-1.

### 2.4. Ethical Issues

This study was conducted in accordance with the Declaration of Helsinki and the principles of Good Clinical Practice. Written consent was obtained from each subject enrolled in the study and the study protocol was approved by The Bioethics Committee of Medical University of Lodz (RNN/176/18/KE).

### 2.5. Data Analysis

Normality of data distribution was checked using Shapiro–Wilk *W* test. The homogeneity of variance was tested by Brown–Forsythe modification of Levene test. One-way ANOVA and Kruskal–Wallis tests were used to compare difference between groups. Then, post hoc procedures were applied to determine significance of differences between specific groups: pairwise comparisons of group II or III against group I were performed by multiple comparisons using the two-sided Dunnett’s *t*-test and contrasts were applied to compare groups II and III. The Bonferroni–Dunn test was used for post hoc analysis after Kruskal–Wallis test. All statistical analyses were performed with STATISTICA 13.3 software (TIBCO Software Inc., Palo Alto, CA, USA).

## 3. Results

General characteristics of the individuals enrolled in this study and the results of routine laboratory tests are presented in Table 1.

The average daily intake of L-tryptophan in group III was significantly (*p* < 0.001) lower than that in groups I and II. Patients in group III showed significantly (*p* < 0.001) higher scores of the Hamilton Depression Rating Scale than those in groups I and II.

A substantial proportion of elderly patients (groups II and III) suffered from somatic diseases that are presented in Table 2.

No significant differences were observed in the occurrence of somatic diseases between groups II and III. Due to these diseases, elderly patients took appropriate medications according to pre-established recommendations (Table 3).

No significant differences were observed in the drug usage between these two groups.

During the 21-day follow-up, patients measured their blood pressure twice a day and diabetic patients measured their blood glucose levels. These values were stable and no dose adjustment was necessary.

The urinary levels of TRP in group III was lower than that in younger adults: 10.4 ± 1.18 vs. 13.3 ± 2.31 mg/gCr (*p* < 0.05, Figure 1A), but there was no difference in the levels of urinary 5-hydroxyindoleacetic acid (5-HIAA, Figure 1B) and TRP/5-HIAA ratio (Figure 1C) between these groups. As 5-HIAA is a main TRP metabolite in its serotonin metabolic pathway, we speculated that this pathway was not affected in older adults with mood disorders as these subjects presented the same 5-HIAA/TRP ratio as the remaining two groups.

The level of L-kynurenine (KYN) in group III was higher than that in group I: 0.85 ± 0.21 vs. 0.45 ± 0.09 mg/gCr (*p* < 0.001, Figure 2B). However, the KYN/TRP ratio in group III was significantly higher than that in the control group: 0.08 ± 0.02 vs. 0.03 ± 0.01 mg/gCr (*p* < 0.001, Figure 2C). As KYN is the main metabolite of TRP in its kynurenine pathway, we reasoned that this pathway was potentiated in older adults with mood disorders. This was confirmed by the increased ratio of KYN to TRP in these patients (Figure 2C).

Elderly individuals with mood disorders showed a higher level of kynurenic acid in urine than subjects in groups II and I (2.93 ± 0.92 mg/gCr vs. 2.20 ± 0.63 mg/gCr and 2.08 ± 0.47 mg/gCr, respectively, *p* < 0.001, Figure 3A). Similar differences occurred in the levels of xanthurenic acid: group I, 0.73 ± 0.27 mg/gCr; group II, 0.84 ± 0.28 mg/gCr; group III, 1.00 ± 0.32 mg/gCr (*p* < 0.001, Figure 3B). Older adult individuals showed a significantly increased (*p* < 0.001) level of quinolinic acid (7.13 ± 1.03 mg/gCr), whereas the level of QA in group II was 4.18 ± 1.19 mg/gCr and in group I was 3.10 ± 1.05 mg/gCr (Figure 3C). These results supported those presented in Figure 2 (the kynurenine pathway of TRP metabolism was potentiated in elderly with mood disorders as the concentration of products of this pathway was increased in these subjects. The ratio of KynA, XA, and QA to TRP was also increased in that group.

Finally, we compared the concentration of tryptophan and its metabolites as well as the metabolite ratios in the three groups with dependence on gender, as the total number of women enrolled in our study (*n* = 60) was significantly higher than men (*n* = 30).

In group II, women presented a higher KYN concentration and 5-HIAA/KYN ratio than men (Table 4, *p* < 0.05 in both cases). In group III, women had lower XA concentration than men (*p* < 0.05). No difference was observed between women and men in the remaining parameters in any group.

## 4. Discussion

The process of aging is associated with a loss of functional reserve of multiple organ systems, increased prevalence of chronic diseases, and enhanced susceptibility to stress [14]. These consequences of aging occur at a different step and with a different rate in individuals, resulting in a great heterogeneity within the elderly. It is still a challenging task to answer the question on the reasons for the difference between chronological and biological aging in various individuals [15].

Mood disorders are relatively frequent in older adults and become a major public health problem [16]. Depression in the elderly may be associated with medical comorbidities and cognitive decline, in addition to increased risk of dementia, suicide attempts, and overall mortality [17].

There are differences in clinical presentation and pathogenesis of mood disorders in older adults and nutritional neuroscience is an emerging branch indicating that nutrition is related to cognition, behavior, and emotions [18]. Therefore, diet may have clinical implications and is important for the effective treatment of mood disorders in the elderly.

In the present work, we found that older adults with mood disorders had a lower intake of tryptophan than their peers and younger individuals without psychiatric problems. As tryptophan cannot be synthesized by humans, its administration is one of the main determinants of its fate in human body. The other is the activity of enzymes involved in TRP metabolism. We showed that individuals with mood disorders consumed less TRP with their diet than their peers without mood disorders and younger individuals. Further, we showed that the concentration of products of a major TRP metabolic pathway was increased in the elderly subjects with mood disorders, independently of whether it was expressed directly or related to TRP amount. Therefore, the subjects with mood disorders might display increased activity of enzymes of the kynurenine pathway of TRP metabolism.

It was not surprising that subjects with mood disorders had lower levels of TRP as compared with individuals of the two remaining groups, as they were characterized by lower intake of TRP. However, such diminished administration of tryptophan in individuals with mood disorders did not hamper the increase in the production of metabolites of the kynurenine pathway of TRP degradation. In fact, over 95% of free TRP is a substrate for this pathway in normal subjects [19]. TRP metabolism through this pathway is mainly involved in the regulation of immunological response, intestinal homeostasis, and neuronal functions [20].

Subjects with depression had lower urine levels of TRP and 5-HIAA and lower 5-HIAA/TRP ratio, which suggested lower tryptophan hydroxylase activity in these subjects. On the other hand, the levels of KYN and the KYN/TRP ratio were increased, suggesting an increased IDO activity.

Decreased levels of TRP was observed in the blood of patients with depression in several clinical trials [21,22,23]. Although clinical trials mainly concentrate on supplementing or depriving TRP or its metabolites for the treatment of neuropsychiatric diseases, current preclinical efforts in drug development for these diseases have mainly focused on altering the rheostat of neuroactive metabolites of the kynurenine pathway [24]. In humans, the KYN/TRP ratio, revealing the involvement of the KYN pathway, increases with age [25]. Such increase was also reported in patients with depression, but other studies showed no association or even a decrease [26,27,28]. Despite these somehow controversial results, TRP supplementation is considered potentially beneficial in many neurological and psychiatric diseases [24]. However, the optimal dose of tryptophan in the prevention or treatment of specific diseases is yet to be established. It is accepted that a daily dose of tryptophan at 5.0 mg/kg of body weight is sufficient for the basic needs of the normal organism, but aging may increase this dose.

Although we enrolled a significantly higher number of women (*n* = 60) than men (*n* = 30) (Table 1), there were no differences between the women to men ratio in each group. That is why our analysis did not include gender as a confounder. However, we made some calculations to check whether the parameters we investigated differed in women and men (Table 4). For seven parameters in three groups, making a total of twenty-one quantities, we observed a gender-specific difference only in three cases. Therefore, the kynurenine tryptophan metabolic pathways may not be strongly gender-dependent.

Our study had several limitations, which point at important elements of further research. Firstly, the number of subjects enrolled was not very impressive, but we had relatively homogenous and well characterized groups. The mood disorders were diagnosed on the basis of Hamilton Depression Scale and no further psychiatric characteristics of the subject were determined. The data on tryptophan intake were taken from information provided by subjects, who recorded the kind and amount of food they consumed. We assumed that the provided information was honest and reliable. It was the input for food calculator that gave data on the TRP content in the meals that the subjects had. We performed our analysis in urine adding at least one step to metabolic changes of TRP in blood.

In conclusion, older adults with mood disorders consumed less tryptophan than their peers without mental disturbances. The elderly with mood disorders were also characterized by a potentiated kynurenine pathway of the tryptophan metabolism. Therefore, further research should determine whether diet supplementation with tryptophan may be beneficial in the prevention and treatment of mood disorders in the elderly. Further studies on the role of enzymes of the kynurenine pathway in the pathogenesis of mood disorders may assess their potential as a target in the treatment of such disorders in the elderly.

## Figures and Tables

**Figure 1 nutrients-12-03183-f001:**
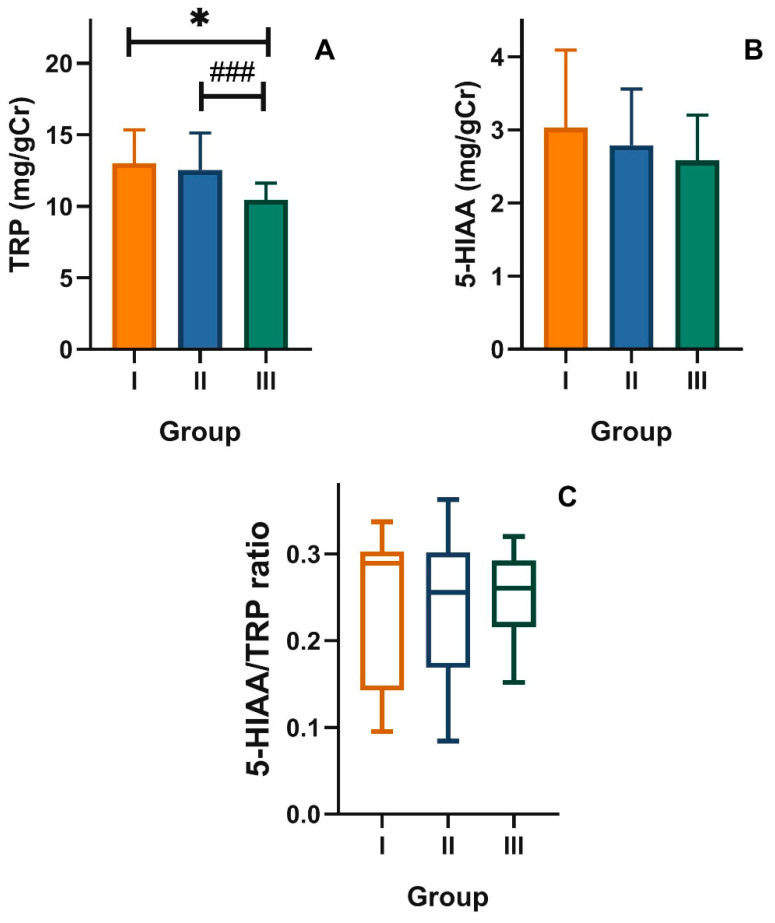
Urinary levels of (**A**) tryptophan (TRP) and (**B**) 5-hydroxyaminoacetic acid (5-HIAA) expressed in milligram per gram of creatinine (mg/gCr), and (**C**) 5-HIAA/TRP ratio in healthy young adult individuals (group I) and in the elderly without (group II) and with mood disorders (group III); mean ± SD (**A**,**B**) or median with boxes represent I and III quartiles, and error bars represent 1.5 times the interquartile distance. Differences between groups were analyzed by ANOVA (**A**,**B**) with Dunnett’s multiple comparison method (group III vs. I) or contrast (III vs. II); the differences between groups in C were assessed by Kruskal–Wallis test; *n* = 30 in each group; * *p* < 0.05; ^###^
*p* < 0.001.

**Figure 2 nutrients-12-03183-f002:**
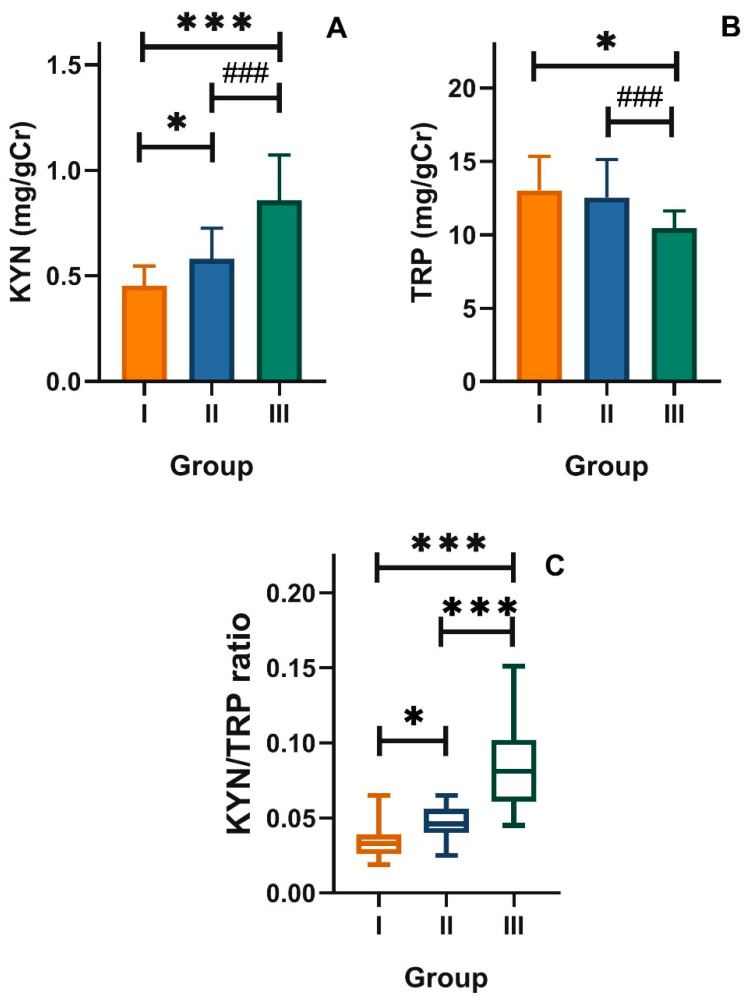
Urinary levels of (**A**) L-kynurenine (KYN) and (**B**) tryptophan (TRP) expressed in milligram per gram of creatinine (mg/gCr), and (**C**) kynurenine/tryptophan ratio (KYN/TRP) in healthy young adult individuals (group I) and in the elderly without (group II) and with mood disorders (group III); mean ± SD (**A**,**B**) or median with boxes represent I and III quartiles, and error bars represent 1.5 times the interquartile distance. Differences between groups were analyzed by ANOVA (**A**,**B**) with Dunnett’s multiple comparison method (group III vs. I and group II vs. I) or contrast (group III vs. II); the differences between groups in C were assessed by Kruskal–Wallis test; *n* = 30 in each group; * *p* < 0.05, *** *p* < 0.001; ^###^
*p* < 0.001.

**Figure 3 nutrients-12-03183-f003:**
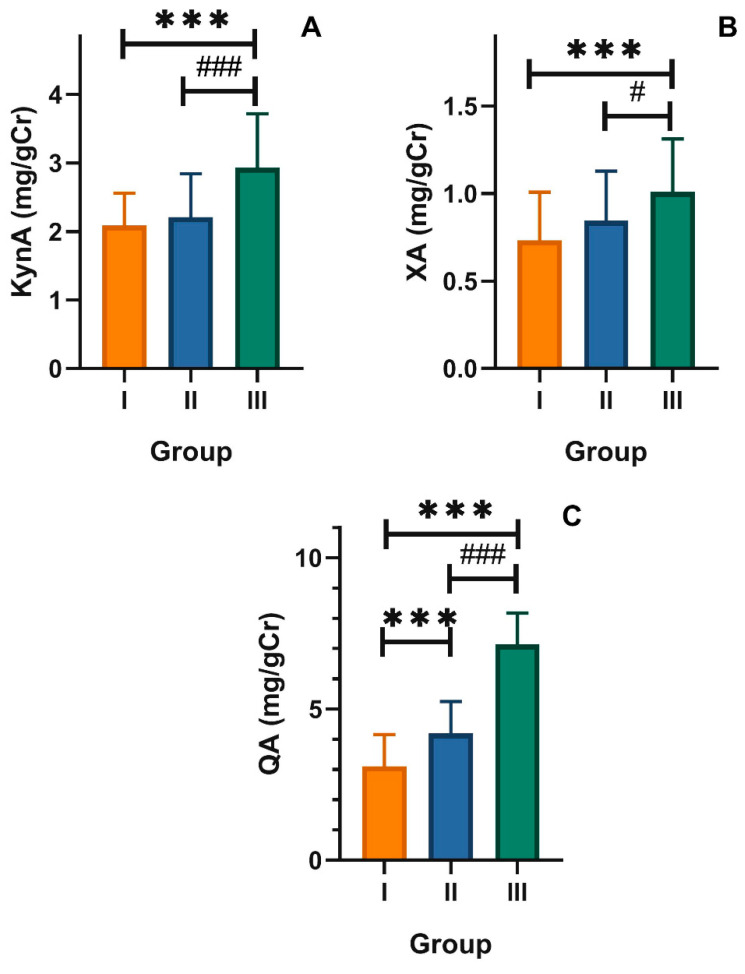
Urinary levels of (**A**) kynurenic acid (KynA), (**B**) xanthurenic acid (XA), and (**C**) quinolinic acid (QA) expressed in milligram per gram of creatinine (mg/gCr) in healthy young adult individuals (group I) and in the elderly without (group II) and with mood disorders (group III); mean ± SD differences between groups were analyzed by ANOVA with Dunnett’s multiple comparison method (group III vs. I and group II vs. I) or contrast (group III vs. II); *n* = 30 in each group; *** *p* < 0.001; ^#^
*p* < 0.05; ^###^
*p* < 0.001.

**Table 1 nutrients-12-03183-t001:** Characteristics of the subjects enrolled in this study: group I, controls; group II, elderly subjects; group III, elderly patients with mood disorders.

Feature ^a^	Group I (*n* = 30)	Group II (*n* = 30)	Group III (*n* = 30)	*p*
Age (years)	42.1 ± 4.2	75.3 ± 4.7	74.6 ± 5.1	ns
Gender				
M	12	8	10	ns
F	18	22	20	ns
BMI (kg/m^2^)	24.8 ± 3.2	26.1 ± 2.4	25.4 ± 3.1	ns
GFR (mL/min)	92 ± 11.2	78.2 ± 14.3	82.6 ± 20.1	ns
ALT (µ/L)	16.5 ± 3.4)	16.4 ± 1.9	17.2 ± 5.4	ns
AST (µ/L)	12.1 ± 2.6	18.2 ± 3.3	20.9 ± 6.1	ns
CRP (µ/g)	0.99 ± 0.63	1.14 ± 0.86	1.65 ± 1.02	ns
TRP daily intake (mg)	1446 ± 201	1370 ± 242	826 ± 106	<0.01 *^,#^
HAM-D score	5.1 ± 1.2	4.9 ± 0.9	13.6 ± 3.7	<0.001 *^,#^

^a^ average ± SD; SD, standard deviation; M, male; F, female; BMI, body mass index; GFR, glomerular filtration rate; ALT, alanine aminotransferase; AST, asparagine aminotransferase; CRP, C-reactive protein; TRP, L-tryptophan; HAM-D, Hamilton Depression Rating Scale; ns (non-significant), *p* > 0.05; * group I vs. group III; ^#^ group II vs. group III.

**Table 2 nutrients-12-03183-t002:** Somatic diseases in elderly patients without (group II) and with (group III) mood disorders.

Disease	Group II (*n* (%))	Group III (*n* (%))
Hypertension	12 (40.0)	16 (53.3)
Coronary disease	7 (23.3)	6 (20.0)
Diabetes	8 (26.6)	9 (30.0)
Dyslipidemia	11 (37.8)	17 (56.6)
Bowel disorders	16 (53.3)	16 (53.3)

**Table 3 nutrients-12-03183-t003:** Drugs used by elderly patients without (group II) and with (group III) depressive mood disorders.

Drugs	Group II (*n* (%))	Group III (*n* (%))
Beta-blockers	11 (36.6)	9 (30.0)
Calcium channel blockers	6 (20.0)	10 (33.3)
Angiotensin inhibitors	12 (40.0)	8 (26.6)
Sartans	6 (20.0)	4 (13.3)
Statins	11 (36.6)	12 (40.0)
Anticoagulant drugs	9 (30.0)	11 (36.6)
Antidiabetic drugs	8 (26.6)	9 (30.0)
Other	19 (63.3)	18 (60.0)

**Table 4 nutrients-12-03183-t004:** Urinary levels of tryptophan (TRP) and its metabolites expressed in milligram per gram of creatinine and their ratios in healthy young adult individuals (group I) and in the elderly without (group II) and with mood disorders (group III) ^1^.

Group	I		II		III	
Gender	M	F	M	F	M	F
TRP	13.4 ± 2.54	12.78 ± 2.18	12.29 ± 1.96	12.64 ± (2.81	10.26 ± 1.48	10.54 ± 1.04
5-HIAA	2.68 ± 0.86	3.26 ± 1.12	2.86 ± 0.98	2.76 ± 0.7	2.37 ± 0.67	2.68 ± 0.59
KYN	0.43 ± 0.09	0.47 ± 0.09	0.49 ± 0.11	0.61 ± 0.14 *	0.91 ± 0.25	0.83 ± 0.2
5-HIAA/TRP	0.19 (0.13–0.30)	0.29 (0.16–0.3)	0.26 (0.19–0.3)	0.25 (0.17–0.3)	0.24 (0.2–0.28)	0.27 (0.22–0.29)
5-HIAA/KYN	0.03 (0.02–0.04)	0.03 (0.03–0.04)	0.04 (0.03–0.04)	0.05 (0.04–0.06) *	0.09 (0.07–0.13)	0.07 (0.06–0.1)
KynA	2.01 ± 0.5	2.13 ± 0.46	2.36 ± 0.72	2.15 ± 0.61	2.97 ± 0.9	2.91 ± 0.74
XA	0.85 ± 0.23	0.65 ± 0.28	0.68 ± 0.17	0.9 ± 0.29	1.17 ± 0.20	0.92 ± 0.31 *
QA	3.15 ± 0.79	3.07 ± 1.22	4.22 ± 0.45	4.18 ± 1.21	6.70 ± 0.64	7.34 ± 1.14

^1^ Mean ± SD or median and the range of I and III quartiles; 5-HIA, 5-hydroxyaminoacetic acid; KYN, L-kynurenine; KynA, kynurenic acid; XA, xanthurenic acid; QA, quinolinic acid; M, men; F, women; * *p* < 0.05 as compared with men.

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
