# Peer review of "Tryptophan Intake and Metabolism in Older Adults with Mood Disorders"

_nutrients, 2020, doi:10.3390/nu12103183_

Round 1
Reviewer 1 Report
This is a study that aims to investigate a possible relationshop between intake and metabolism of tryptophan in elderly subjects with mood disorders.
The study is interesting and well presented, although it suffers from some limitations, as admitted by the authors themselves, such as the poor number of subjects enrolled.
As shown in Materials and Methods, as regards the gender of the subjects enrolled, it is noted that women are more than twice as many as men.
Even within groups (controls, elderly subjects, and elderly patients with mood disorders) women enrolled are many more.
It would be very interesting to provide some statistical analysis separate by gender and investigate possible correlations in order evaluate if there are significant differences in the results between males and females.
Discussion:
lines 253 -256
“Therefore, the diet of the elderly may be supplemented with tryptophan to prevent their mood disorders. Tryptophan can be also used as a drug in mood disorders in older adults. The enzymes of the kynurenine pathway can be targeted in mood disorders therapy”.
I think these final sentences are too strong and bold and need to be revised.
The study demonstrate the association of of mood disorders with tryptophan intake and metabolism in the elderly, but not that increasing tryptophan intake has the power to prevent mood disorders or that it can even be used “as a drug".
Also the sentence “the enzymes of the kynurenine pathway can be targeted in mood disorders therapy” is not really supported by the results.
Maybe these sentences should be removed and replaced with more cautious conclusions.
Author Response
Reviewer #1
This is a study that aims to investigate a possible relationshop between intake and metabolism of tryptophan in elderly subjects with mood disorders.
The study is interesting and well presented, although it suffers from some limitations, as admitted by the authors themselves, such as the poor number of subjects enrolled.
Comment: As shown in Materials and Methods, as regards the gender of the subjects enrolled, it is noted that women are more than twice as many as men.
Even within groups (controls, elderly subjects, and elderly patients with mood disorders) women enrolled are many more.
Answers: We have added the following fragment to the Discussion section:
“Although we enrolled a significantly higher number of women (60) than men (30) (Table 1), there were no differences between the women to men ratio in each group. That is why our analysis did not include gender as a confounder. However, we made some calculations to check whether the parameters we investigated differed in women and men (Table 4). For 7 parameters in 3 groups, making a total of 21 quantities, we observed a gender-specific difference only in 3 cases. Therefore, the kynurenine tryptophan metabolic pathways is not strongly gender-dependent.”
Comment: It would be very interesting to provide some statistical analysis separate by gender and investigate possible correlations in order evaluate if there are significant differences in the results between males and females.
Answer: We have added the following fragment to the end of the Results section:
“Finally, we compared the concentration of tryptophan and its metabolites as well as the metabolite ratios in the three groups with dependence on gender as the total number of women enrolled in our study (60) was significantly higher than men (30).
Table. 4. Urinary levels of tryptophan (TRP) and its metabolites expressed in milligram per gram of creatinine and their ratios in healthy young adult individuals (group I), and in the elderly without (group II), and with mood disorders (group III); mean ± SD or median and the range of I and III quartiles1.
|
Group |
I |
II |
III |
|||
|
Gender |
M |
F |
M |
F |
M |
F |
|
TRP |
13.4 ± 2.54 |
12.78 ± 2.18 |
12.29 ± 1.96 |
12.64 ± (2.81 |
10.26 ± 1.48 |
10.54 ± 1.04 |
|
5-HIAA |
2.68 ± 0.86 |
3.26 ± 1.12 |
2.86 ± 0.98 |
2.76 ± 0.7 |
2.37 ± 0.67 |
2.68 ± 0.59 |
|
KYN |
0.43 ± 0.09 |
0.47 ± 0.09 |
0.49 ± 0.11 |
0.61 ± 0.14* |
0.91 ± 0.25 |
0.83 ± 0.2 |
|
5-HIAA/TRP |
0.19 (0.13-0.30) |
0.29 (0.16-0.3) |
0.26 (0.19-0.3) |
0.25 (0.17-0.3) |
0.24 (0.2-0.28) |
0.27 (0.22-0.29) |
|
5-HIAA/KYN |
0.03 (0.02-0.04) |
0.03 (0.03-0.04) |
0.04 (0.03-0.04) |
0.05 (0.04-0.06)* |
0.09 (0.07-0.13) |
0.07 (0.06-0.1) |
|
KynA |
2.01 ± 0.5 |
2.13 ± 0.46 |
2.36 ± 0.72 |
2.15 ± 0.61 |
2.97 ± 0.9 |
2.91 ± 0.74 |
|
XA |
0.85 ± 0.23 |
0.65 ± 0.28 |
0.68 ± 0.17 |
0.9 ± 0.29 |
1.17 ± 0.20 |
0.92 ± 0.31* |
|
QA |
3.15 ± 0.79 |
3.07 ± 1.22 |
4.22 ± 0.45 |
4.18 ± 1.21 |
6.70 ± 0.64 |
7.34 ± 1.14 |
15-HIA – 5-hydroxyaminoacetic acid, KYN – L-kynurenine, KynA – kyneurenic acid, XA – xanthurenic acid, QA – quinolinic acid , M – men, F – women, * – p < 0.05 as compared with men.
In group II, women presented a higher KYN concentration and 5-HIAA/KYN ratio than men (Table 4, p < 0.05 in both cases). In group III, women had lower XA concentration than men (p < 0.05). No difference was observed between women and men in the remaining parameters in any group.
Comment: Discussion:
lines 253 -256
“Therefore, the diet of the elderly may be supplemented with tryptophan to prevent their mood disorders. Tryptophan can be also used as a drug in mood disorders in older adults. The enzymes of the kynurenine pathway can be targeted in mood disorders therapy”.
I think these final sentences are too strong and bold and need to be revised.
The study demonstrate the association of of mood disorders with tryptophan intake and metabolism in the elderly, but not that increasing tryptophan intake has the power to prevent mood disorders or that it can even be used “as a drug".
Also the sentence “the enzymes of the kynurenine pathway can be targeted in mood disorders therapy” is not really supported by the results.
Maybe these sentences should be removed and replaced with more cautious conclusions.
Answer: This is completely right! We overinterpreted and overestimated the results we obtained. We have changed the fragment:
“Therefore, the diet of the elderly may be supplemented with tryptophan to prevent their mood disorders. Tryptophan can be also used as a drug in mood disorders in older adults. The enzymes of the kynurenine pathway can be targeted in mood disorders therapy”.
into:
“Therefore, further research should determine whether diet supplementation with tryptophan may be beneficial in the prevention and therapy of mood disorders in the elderly. Further studies on the role of enzymes of the kynurenine pathway in the pathogenesis of mood disorders may asses their potential as a target in the treatment of such disorders in the elderly.”
Reviewer 2 Report
The paper is an interesting study exploring the tryptophan (TRP) intake and metabolism in elderly mood disorders patients, on the basis of the serotonin hypothesis of depression. The strength and originality of the study is that the authors assessed both TRP intake and the urinary metabolites of the TRP shunt. The findings highlighted low TRP intake coupled with increased urinary concentrations of 5-hydroxyindoleacetic acid, L-kynurenine ), kynurenic acid , xanthurenic acid, and quinolinic acid, while suggesting that tRP supplementation might be useful in these kind of depressed patients.
The size of the sample is adeguate for the statistical analyses employed. The results are clear and the consequent discussion congruent with the overall findings and with reference to available literature.
Before the publication, the authors should answer the following questions
- Why you did not compare young vs elderly patients
- More stress on the use of peripheral, i.e., urinary serotonergic markers should be given
- The English language should eb slightly revised by a Native English speaker
Author Response
The paper is an interesting study exploring the tryptophan (TRP) intake and metabolism in elderly mood disorders patients, on the basis of the serotonin hypothesis of depression. The strength and originality of the study is that the authors assessed both TRP intake and the urinary metabolites of the TRP shunt. The findings highlighted low TRP intake coupled with increased urinary concentrations of 5-hydroxyindoleacetic acid, L-kynurenine ), kynurenic acid , xanthurenic acid, and quinolinic acid, while suggesting that tRP supplementation might be useful in these kind of depressed patients.
The size of the sample is adeguate for the statistical analyses employed. The results are clear and the consequent discussion congruent with the overall findings and with reference to available literature.
Before the publication, the authors should answer the following questions
Comment: Why you did not compare young vs elderly patients
Answer: All possible comparisons, including group I vs. II, have been done – please see Figures 1-3.
Comment: More stress on the use of peripheral, i.e., urinary serotonergic markers should be given
Answer: Our manuscript presents results obtained on the kynurenic pathway of tryptophan metabolism, so it is not justified to write too much about the other, 5-HT, metabolic pathway. Despite that, we have addressed this issue in lines 45-60 and in some places in the Discussion and we really think that this is adequate to the main subject of our manuscript.
Comment: The English language should eb slightly revised by a Native English speaker
Answer: We have done our best to improve the style of the manuscript. Thank you.